# Exploring prompts to elicit memorization in masked language model-based named entity recognition

**Yuxi Xia**[1,2*], **Anastasiia Sedova**[1,2], **Pedro Henrique Luz de Araujo**[1,2], **Vasiliki Kougia**[1,2], **Lisa Nußbaumer**[3], **Benjamin Roth**[1,3]

**1** Faculty of Computer Science, University of Vienna, Vienna, Austria, **2** UniVie Doctoral School Computer Science, Vienna, Austria, **3** Faculty of Philological and Cultural Studies, University of Vienna, Vienna, Austria

\* yuxi.xia@univie.ac.at

**Data availability statement:** All data is held in a public repository named "NER-memorization-detection" (https://github.com/Yuuxii/NER-memorization-detection). And the data are

## Abstract

The possibility of identifying specific information about the training data a language model memorized poses a privacy risk. In this study, we analyze the ability of prompts to detect training data memorization in six masked language models, fine-tuned for named entity recognition. Specifically, we employ a diverse set of 1,200 automatically generated prompts for three entity types and a detection dataset that contains entity names present in the training set (in-sample names) and names not present (out-of-sample names). Here, prompts constitute patterns that can be instantiated with candidate entity names, and the prediction confidence of a corresponding entity name serves as an indicator of memorization strength. The prompt performance of detecting memorization is measured by comparing the confidences of in-sample and out-of-sample names. We show that the performance of different prompts varies by as much as 24.5 percentage points on the same model, and prompt engineering further increases the gap. Moreover, our experiments demonstrate that prompt performance is model-dependent but does generalize across different name sets. We comprehensively analyze how prompt performance is influenced by prompt properties (e.g., length) and contained tokens.

## 1 Introduction

Recent studies have highlighted the memorization of training data in language models [1], especially in auto-regressive ones (e.g., GPT models) [2,3]. Such studies are particularly important in exploring models' privacy risk [1], as attacks like membership inference (MI) [4, 5] can leverage data memorization to extract sensitive training data from the model. While training data can be extracted through direct prompt querying and text generation in GPT models, that method does not apply to Masked Language Models (MLMs).

Despite the efficiency and state-of-the-art performance achieved by MLM-based Named Entity Recognition (NER) models across various domains compared to GPT models [6], there is a lack of work studying memorization in NER models. During fine-tuning for NER, the model's objective shifts entirely to sequence labeling, and the token reconstruction head

all contained within the Supporting information files.

**Funding:** This study has been funded by the Vienna Science and Technology Fund (WWTF)[10.47379/VRG19008] "Knowledge-infused Deep Learning for Natural Language Processing" (https://www.wwtf.at) in the form of a grant, received by YX, BR, PHLdA, and VK. This study was also financially supported by Open Access funding provided by University of Vienna (https://openaccess.univie.ac.at/) in the form of a grant, received by YX.

**Competing interests:** The authors have declared that no competing interests exist.

used during pre-training is replaced with a task-specific classification head. This new output layer predicts only NER tags and no longer retains the capability to generate raw text [7–9]. Hence, NER models cannot be directly assessed for memorization via text generation, such as prompting them to reproduce verbatim in-sample entities given a prefix, a strategy commonly applied to auto-regressive models [3].

This paper explores prompts' impact on detecting the memorization of three types of entity names in six NER models, where prompts refer to linguistic context templates into which named entities are inserted [10–12]. Unlike previous work [13], who used only five hand-written prompts to detect memorization in a self-fine-tuned NER model on private, non-public data, which overlooks the impact of prompts, leading to limitations in robustness. We assess model sensitivity to prompt variations [14]. Specifically, we employ a set of 1,200 diverse prompts generated by GPT-3.5-turbo [15] for *PER* (person), *LOC* (location), and *ORG* (organization) entities. Our prompt set spans four sentence types (declarative, exclamatory, imperative, and interrogative), 15 prompt lengths, and 11 token positions for the target name. To ensure transparency and reproducibility, we evaluate publicly accessible state-of-the-art NER models fine-tuned on the widely-used CoNLL2003 dataset [16].

To quantify model memorization, we create a detection dataset by sampling 5,000 in-sample entity names from the CoNLL-2003 training data and 5,000 out-of-sample names from external sources (Wikidata [17], WikiANN [18]). Fig 1 illustrates our full memorization detection pipeline. Each prompt from the prompt set is completed with all entity names in the detection dataset, and the resulting sentences are input into the NER model to generate confidence scores for each name. These scores are ranked to compute the model memorization (*M-MEM*) score, which corresponds to the AUC score used in membership inference attacks (MIA) [4], thus quantifying the extent of memorization in the model. The *M-MEM* score detected by a prompt is referred to as the performance of this prompt.

Our experimental results demonstrate that memorization detection is sensitive to prompt variations. The performance gap using different prompts on the same model can be as large as 24.5 percentage points, addressing the limitation of using a single hand-written prompt for memorization detection. To provide solid verification of this finding, we perform extensive experiments to answer three research questions:

- **Are all models sensitive to prompt variations?** The average performance gaps over 3 entity types in the prompt set of six studied NERs range from 9.93 to 20.07, demonstrating that all models are sensitive to prompt variations regardless of pertaining schemes and model sizes;

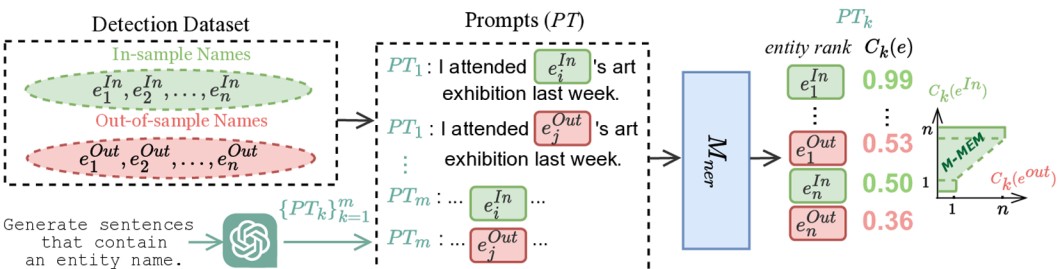

**Fig 1. Memorization detection process of NERs.** Each generated prompt is completed with all entity names from the detection dataset as inputs to the NER model $M_{ner}$. $C_k(e)$ represents the prediction confidence of $M_{ner}$ for an entity name in $PT_k$. The performance of prompt $PT_k$ for detecting memorization is measured by *M-MEM* score.

- **Are prompt performances generalizable?** By computing Kendall's $\tau$ coefficient of prompt performance between different models and name sets, we find that prompt performance is model-dependent but generalizes across mutually exclusive name sets;
- **Can we further increase the prompt performance gap?** Experiments using prompt engineering and different ensembling techniques show that prompt engineering can increase the performance gap up to 51.3 percentage points.

For better guidance in generating higher/lower performing prompts, we perform two different levels of analysis of how various prompt properties impact the performance: (1) Sentence-level analysis of prompt types, length and token positions of the name; (2) Token-level analysis of individual token importance of a prompt. Furthermore, we use self-attention weights to interpret how different prompts change the model's attention weights which leads to performance differences.

Overall, our study quantifies memorization of NER models in a structured prediction task, without relying on models' generative capacity. We are the first to analyze the prompts' influence on the memorization detection of MLM-based NERs, using a diverse set of 1,200 prompts and six publicly accessible NER models for three entity types. Our experimental results strongly prove the importance of prompt choice in the memorization detection of NERs. We recommend that future research utilize diverse prompts (or generate target prompts by following our paper) to detect more robust memorization scores for privacy studies. The data and code used for the paper are provided in S2 Dataset and S3 Code.

## 2 Related work

**Memorization in auto-regressive language models.** This is first studied by Carlini et al. [19], who show that LSTMs memorize a significant fraction of their training data. Later, Carlini et al. [20] and Carlini et al. [3] use extractability to measure the memorization of GPT-2 and GPT-Neo models. Wei et al. [21] show that prefixed prompts can extract sensitive training data. However, measuring memorization with extractability only applies to auto-regressive language models, but not to fine-tuned MLMs. This is because auto-regressive language models can be directly probed for memorization by prompting them to generate text given a prefix. A model that reproduces long sequences in-sample during training, especially verbatim, indicates a strong signal of memorization. Thus, memorization in auto-regressive models is typically studied through generation-based evaluations [3,20,21]. Wang et al. [22] demonstrate the possibility to generate low-quality text from pre-trained BERT based on Gibbs sampling. However, this can not be applied to fine-tuned MLMs, because the token reconstruction head used during pre-training is replaced with a task-specific classification head which is restricted to the target space, e.g., named entity labels [7–9]. This makes standard generation-based memorization probing inapplicable to NERs.

**Memorization in MLMs.** Tirumala et al. [23] and Magar et al. [24] study memorization in MLMs by detecting specific data items. However, they focus more broadly by measuring model generalization using metrics related to the accuracy difference between training and test sets. Ali et al. [13] examine **memorization in fine-tuned MLMs** (NER models) and average the confidence ranks of in-sample entities in an entity set to measure model memorization. However, Ali et al. [13] self-fine-tune the NER model and study the impact of the training data duplication and complexity on the memorization dynamics. Differently, we explore 6 publicly accessible NER models fine-tuned by previous work and focus on the impact of different prompts on memorization detection.

**Hand-written vs. Generated prompts.** Language model performance has been shown to be sensitive to prompt variations in previous work [14,25]. Thus, randomly choosing one of the five hand-written prompts for each studied entity to detect memorization as in Ali et al. [13] may not robustly represent the model memorization. Instead, we employed 1,200 generated diverse prompts and studied the memorization, showing a large variability of memorization values associated with different prompts.

## 3 Methodology

In this section, we first describe the processes for creating the detection dataset and prompt set. Then, we introduce how we use them to detect memorization in NER models.

### 3.1 Detection dataset creation

We create a detection dataset $D_{dt}$ to quantify the memorization of NER models fine-tuned on the CoNLL-2003 dataset [16]. Specifically, we first generate an in-sample name set $D_{In}$ that contains all in-sample names of *PER*, *LOC* and *ORG* from the CoNLL-2003 dataset. Then, we create an out-of-sample name set $D_{Out}$ by gathering names from publicly available data sources–Wikidata [17] and WikiANN [18]. Finally, the detection dataset $D_{dt}$ contains equal number of in-sample and out-of-sample names ($\{e_i^{In}, e_i^{Out}\}_{i=1}^n$) sampled from $D_{In}$ and $D_{Out}$ for each entity type. As there is no training process in our setting, we exclusively split in-sample and out-of-sample names in $D_{dt}$ equally into development (dev) and test. The details of the dataset statistics are shown in Table 1. The full dataset is provided in S1 Dataset.

**In-sample Name Set ($D_{in}$).** CoNLL-2003 is a commonly used NER dataset for research with a large number of examples of named entities. The training dataset of the CoNLL-2003 corpus contains 6,600 *PER* entity names, i.e., entity names with ground-truth label *B-PER* and/or *I-PER* which denote the beginning and the rest of a person's name respectively. After the post-processing of removing the duplicates and single-token *PER*, $D_{in}$ contains 2,645 unique multi-token *PER* entity names. Similarly, we collect the *LOC* entity names by extracting the entity names with ground-truth label *B-LOC* and/or *I-LOC*, and *ORG* entity names with ground-truth label *B-ORG* and/or *I-ORG*. Different from the post-processing of *PER* entity, we delete the duplicates but keep the single-token names for *LOC* and *ORG*, resulting in 1,295 names of *LOC* and 2,297 names of *ORG*.

**Out-of-sample Name Set ($D_{out}$).** Wikidata [17] as a source for *PER* entity names in $D_{out}$ enables the compilation of a list of real-world person names. Specifically, a SPARQL query is formulated to retrieve *PER* entity names. Similar to $D_{in}$, duplicates and single-token *PER* names are filtered out, resulting in 7,617,797 multi-token *PER* names. Additionally, we also removed the *PER* names presented in $D_{in}$, (7,617,797 - 1,651) *PER* names remained in the dataset $D_{out}$ in the end. To simplify the process, the in-sample names of *LOC* and *ORG* are collected from WikiANN [18]. WikiANN is a NER dataset consisting of Wikipedia articles that have been annotated with *LOC*, *PER*, and *ORG* tags. After deleting the duplicates and names presented in $D_{in}$, $D_{out}$ contains 7,266 *LOC* and 10,479 *ORG* names.

Table 1. **Detection dataset statistics.** One name is randomly assigned to dev/test if the total number of names is odd.

| $D_{dt}$ | *PER* | | *LOC* | | *ORG* | |
|---|---|---|---|---|---|---|
| | #In-sample | #Out-of-sample | #In-sample | #Out-of-sample | #In-sample | #Out-of-sample |
| dev | 826 | 826 | 647 | 647 | 1,148 | 1,148 |
| test | 825 | 825 | 648 | 648 | 1,149 | 1,149 |
| total | 1,651 | 1,651 | 1,295 | 1,295 | 2,297 | 2,297 |

**Detection Dataset** ($D_{dt}$)**.** For in-sample *PER* names in $D_{dt}$, we select the ones also presented in Wikidata to ensure a better quality of name selection. This reduces the number of in-sample *PER* names from $D_{in}$ to 1,651. We select all the names of *LOC* and *ORG* in $D_{in}$. Out-of-sample names are randomly sampled from $D_{out}$ for each entity type.

## 3.2 Prompt set generation

To collect a large and diverse set of prompts for each entity type, we use the generative large language model GPT-3.5-turbo-1106 (ChatGPT [15]), to automatically generate *m* diverse prompts $\{PT_k\}_{k=1}^m$, *m* = 400, for each entity type.

To increase the diversity of the prompts, we consider four types of sentences: (1) declarative; (2) exclamatory; (3) imperative; and (4) interrogative. Next, we prompted ChatGPT to generate 100 sentences of each sentence type. The prompt template we used is: *"Generate 100 different [sentence type] sentences that must contain [entity type], and replace the [entity type] with 'MASK' "*. Details about the prompts and generated sentence examples are shown in Table 2. The full prompt set is provided in S1 Dataset.

The *"MASK"* string in a prompt $PT_k$ is completed with $e_i^{In}$ and $e_i^{Out}$ resulting in two separate sentences, which are then fed to NER models to obtain the confidence scores for $e_i^{In}$ and $e_i^{Out}$.

## 3.3 Memorization detection

Let $M_{ner}$ denotes a fine-tuned MLM model for NER task on the CoNLL-2003 dataset. Such models can achieve high classification accuracy in predicting entity labels. When evaluating each token in a target entity, the NER model outputs probabilities over all possible tags (e.g., B-PER and I-PER, which represent the beginning and the inside tags of a person entity).

**Table 2**. **Examples of different types of prompts for different entities.**

| Entity | ChatGPT Prompt | Sentence Types | Auto-Generated Prompts for NER |
|---|---|---|---|
| PER | Generate 100 different [Sentence Type] sentences that must contain a person's name, and replace the person's name with "MASK". | Declarative | MASK is a talented artist. |
| | | | I admire MASK's creativity. |
| | | Exclamatory | Oh, MASK, you're such a genius! |
| | | | Whoa, MASK, that was mind-blowing! |
| | | Imperative | MASK, please pass the salt. |
| | | | MASK, don't forget to lock the door. |
| | | Interrogative | Can MASK pass the message to him? |
| | | | Can you tell me where I can find MASK? |
| LOC | Generate 100 different [Sentence Type] sentences that must contain a location, and replace the location with "MASK". | Declarative | The concert will be held in MASK. |
| | | | We spent our summer vacation in MASK. |
| | | Exclamatory | Wow, I can't believe how beautiful MASK is! |
| | | | The sunsets in MASK are absolutely stunning! |
| | | Imperative | Drive to MASK now. |
| | | | Send the package to MASK. |
| | | Interrogative | Where can I find the best coffee in MASK? |
| | | | What is the population of MASK? |
| ORG | Generate 100 different [Sentence Type] sentences that must contain an organization name, and replace the organization name with "MASK". | Declarative | MASK offers excellent employee benefits. |
| | | | The stock price of MASK surged last week. |
| | | Exclamatory | I can't believe MASK won the award! |
| | | | MASK is revolutionizing the industry! |
| | | Imperative | Attend the MASK meeting next Monday. |
| | | | Donate to MASK to support their cause. |
| | | Interrogative | How do I apply for a job at MASK? |
| | | | What is the mission statement of MASK? |

We extract the maximum probability between the corresponding beginning and inside tags (e.g., B-PER and I-PER) for each token as we insert the target entities and are aware of their correct type (e.g., PER or ORG). This ensures we capture the model's confidence that the token belongs to the correct entity type, regardless of whether it was predicted as a beginning or continuation token. Then, we average these per-token maximum probabilities across all tokens in the entity ($T_e$), yielding a single score that reflects the model's overall confidence $C(e)$ in labeling the full entity with the correct type. We formulate the entity confidence $C(e)$ as:

$$C(e) = \sum_{t \in T_e} \frac{\max(P(B\text{-}e|t), P(I\text{-}e|t))}{|T_e|} \tag{1}$$

Where $B\text{-}e$ and $I\text{-}e$ denote the beginning and inside tags of an entity. We define the confidence of an entity name in $PT_k$ as $C_k(e)$.

**Definition 3.1.** **NER Memorization:** We define NER memorization as the extent to which a NER model exposes training data through its predictions. Specifically, it reflects the likelihood that an entity name from the training set can be identified via a membership inference attack. We quantify memorization as the percentage of instances in which a model assigns higher confidence to an in-sample entity name compared to an out-of-sample counterpart when both are presented in the same prompt.

Assume an entity type in dataset $D_{dt}$ contains $n$ in-sample names $\{e_i^{In}\}_{i=1}^n$ and $n$ out-of-sample names $\{e_j^{Out}\}_{i=1}^n$, we formulate the performance of a prompt $PT_k$ on detecting memorization of $M_{ner}$ for this entity type as $M\text{-}MEM_k$ score:

$$M\text{-}MEM_k = \sum_{i=1}^{n} \sum_{j=1}^{n} \frac{|\{C_k(e_i^{In}) > C_k(e_j^{out})\}|}{n^2} \tag{2}$$

Note that this formulation is equivalent to the AUC metric used for membership inference attack in [4]. It can be interpreted as the average rank of an in-sample name in the out-of-sample name set, where 100% means the in-sample name ranked the top among all out-of-sample names, 0% means the opposite, and 50% is considered as no significant memorization detected.

**Sensitivity and robustness in memorization detection.** To measure the model's sensitivity to prompt variations, we calculate the performance gap ($\Delta$) between the highest and lowest performance across different prompts. The model's robustness is assessed by computing the standard deviation ($\sigma$) of the prompts' performances across the entire prompt set:

$$\sigma_{M_{ner}} = \sqrt{\frac{1}{m} \sum_{k=1}^{m} (M\text{-}MEM_k - \mu)^2} \tag{3}$$

Where $\mu$ is the average value of prompt performances. A lower $\sigma$ value indicates greater robustness to prompt variation.

## 4 Experimental setup

We performed our model memorization analysis on publicly accessible MLM-based NER models fine-tuned by other researchers on the CoNLL-2003 dataset [16]. To provide a more comprehensive and diverse analysis, we selected six models built on MLMs across

three different pretraining schemes (ALBERT-v2 [26], BERT [27], and RoBERTa [28]) and 2 model sizes (base and large sizes, *-B* and *-L* notations are used correspondingly) for each scheme. More details about the models and accessibility information can be found in S1 Appendix.

## 4.1 Strategies of using the prompt set

In addition to the baseline, we investigate three strategies for exploiting the prompt set.

**Baselines (BSL).** $\varnothing$-**PT** uses the entity name without any additional text as the input to query corresponding confidences for memorization detection. One-PT and Mix-PT use the same strategies as in [13]: **One-PT** employs one hand-written prompt (e.g., *"My name is XX."* for *PER* entity), corresponding to the *known*-setting in [13], while **Mix-PT** randomly chooses for each name one of the 5 hand-written prompts, like the *unknown*-setting in [13]. More prompt details are shown in Table 3.

**Original Prompt (OPT)** selects the best prompt (**B-PT**) and worst prompt (**W-PT**) from the prompt set that achieved the highest and lowest *M-MEM* scores on the dev set of $D_{dt}$ for each entity type.

**Prompt Engineering (PTE)** is inspired by [29] who apply a token-removal operation to investigate how tokens of the input influence the model prediction. Here, the goal is to maximize the *M-MEM* score gap of the original best and worst prompts. Specifically, the most important token (which contributes the most to score improvement) and the least important token are removed from the worst and best prompts, respectively, resulting in two modified prompts. This process is iteratively repeated for the modified prompts until only one token (excluding the entity name) remains. The two modified prompts that achieved the highest and lowest *M-MEM* scores on the dev set among all the modified prompts are named **BM-PT** (best-modified prompt) and **WM-PT** (worst-modified prompt). It is important to note that such prompts may be ungrammatical. More details are presented in Sect 5.5.

**Ensembling of Prompts (EPT)** employs several ensemble techniques: (1) Majority Voting (**MV**) decides the rank between an in-sample and an out-of-sample names by majority agreement of the prompt set. (2) Average Confidence (**AVG-C**) uses the average entity confidence of all prompts $avg(\{C_k(e)\}_{k=1}^{k=m})$ as the final confidence score of the entity; (3) Weighted Confidence (**WED-C**) weights the entity confidence of each prompt by its *M-MEM* score; (4) Maximum Confidence (**MAX-C**) denotes the maximum confidence of the entity over all prompts $(max(\{C_k(e)\}_{k=1}^{k=m}))$ as the final confidence; (5) Minimum Confidence score (**MIN-C**) is the contrast to MAX-C, which uses the minimum value $min(\{C_k(e)\}_{k=1}^{k=m})$ as the final confidence.

**Table 3. Hand-written prompts for different entities as baselines.**

| Baseline | Prompt | | |
|---|---|---|---|
| | *PER* | *LOC* | *ORG* |
| $\varnothing$-PT | - | - | - |
| One-PT | My name is MASK. | I am at MASK. | I work for MASK. |
| Mix-PT | My name is MASK. | I am at MASK. | I work for MASK. |
| | I am MASK. | I like MASK. | I like MASK. |
| | I am named MASK. | MASK is a good place. | MASK is a good organization. |
| | Here is my name: MASK. | Meet at MASK. | See you in MASK. |
| | Call me MASK. | Do you live in MASK? | Do you know MASK? |

### 4.2 Interpretability method

**Self-attention weights** [30] of a model can show where the model attends in the sequence and provide insights into what affected the predictions. Specifically, we first average the extracted attention weights of the tokens that correspond to the entity names for all completed sentences of a prompt (e.g., the best prompt completed with all in-sample entity names). Then we create a heatmap using the averaged values over the attention heads and the layers.

## 5 Results and analysis

Results showed in Table 4 demonstrate that the *M-MEM* scores vary considerably depending on the prompt: the maximum performance difference between the best and the worst prompt for a given model is as large as 24.5 percentage points for ALBERT-B on the test set for *LOC* entity.

We validate the statistical significance of the results through Cochran's $Q$ test and found that the differences are significant for all models ($p < 0.001$). For an in-depth investigation, the following sections answer three research questions and perform sentence- and token-level analysis of how different factors impact the prompt performance. Furthermore, we use self-attention weights to interpret how different prompts change the model's attention weights, which leads to the performance differences. Note that we emphasize the *PER* results and analysis because this is the only entity type for which privacy has a special legal status reflected in regulation. E.g., the GDPR institutes a "right to be forgotten" that only applies to persons [31].

### 5.1 Are all models sensitive to prompt variations?

The last row of Table 4 reports the average performance difference across three entity types for the 6 studied NER models. Notably, RoBERTa-L shows the highest robustness, with a standard deviation ($\sigma$) of 1.51, while BERT-L is the least robust, with a $\sigma$ of 2.47. The scores of different models range from 9.93 to 20.07, indicating that **all models are sensitive to prompt variations**, regardless of their pretraining strategies or model sizes.

### 5.2 Are prompt performances generalizable?

To examine how prompt performance generalizes across models and entities, we compute the Kendall's $\tau$ coefficient among the *M-MEM* scores of all the prompts for different models and data splits and show the results of *PER* entity in Fig 2. Results of other entity types are shown in S1 Appendix.

**Generalizability across models:** We observe that *M-MEM* scores on the dev set for BERT-B and BERT-L are negatively correlated, demonstrating a weak correlation between prompts' *M-MEM* scores for different models even when comparing different variants of the same

**Table 4. Sensitivity (Δ) and robustness (σ) of models.** The results are measured with the original prompt set on dev set. Bold/italic highlights the highest/lowest average Δ and σ values.

|  | ALBERT-B | | ALBERT-L | | BERT-B | | BERT-L | | RoBERTa-B | | RoBERTa-L | |
|---|---|---|---|---|---|---|---|---|---|---|---|---|
|  | Δ | σ | Δ | σ | Δ | σ | Δ | σ | Δ | σ | Δ | σ |
| *PER* | 15.92 | 1.63 | 8.52 | 1.66 | 8.85 | 1.52 | 3.78 | 0.55 | 13.76 | 3.15 | 4.66 | 0.79 |
| *LOC* | 24.46 | 3.10 | 14.98 | 2.76 | 7.46 | 1.35 | 15.07 | 3.53 | 12.66 | 1.16 | 9.22 | 0.48 |
| *ORG* | 19.83 | 2.62 | 18.66 | 1.94 | 14.49 | 2.87 | 13.20 | 3.32 | 20.26 | 2.62 | 15.91 | 3.26 |
| *Avg.* | **20.07** | 2.45 | 14.05 | 2.12 | 10.27 | 1.91 | 10.68 | **2.47** | 15.56 | 2.31 | *9.93* | *1.51* |

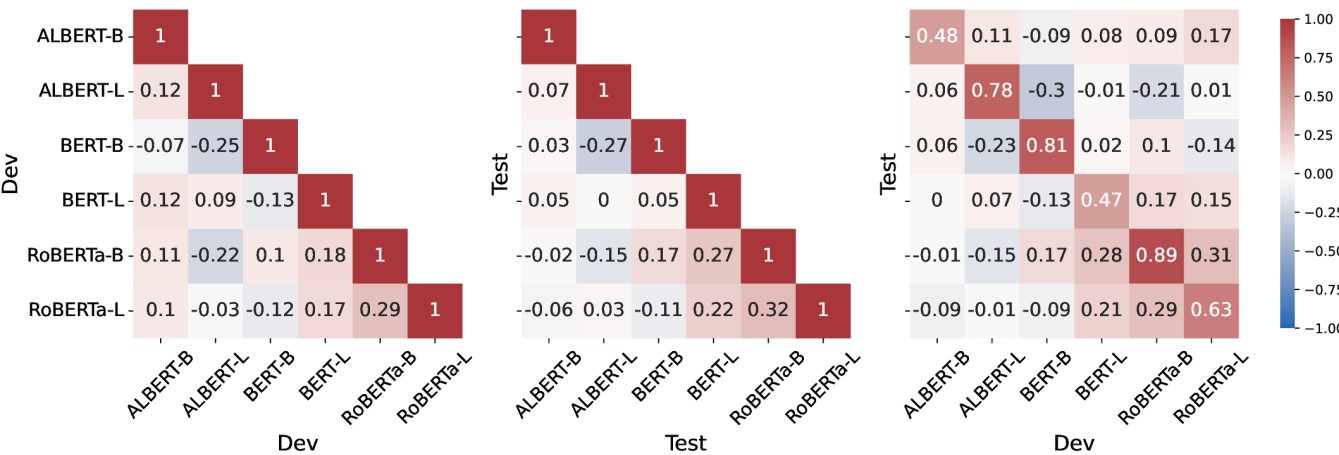

**Fig 2. Correlations of the *M-MEM* scores of prompts across models on *PER* entity.** Left: correlations (Kendall's $\tau$) for the dev set scores. Middle: correlations for the test set scores. Right: correlations between dev and test set scores.

model on the same data split. From this, we conclude that the *M-MEM* scores of prompts do not generalize across models—the best prompts for a given model may not be the best for other models.

**Generalizability across name sets:** Conversely, we found higher correlations between prompts' *M-MEM* scores across data splits when using the same model (diagonal of the right plot in Fig 2). We conclude that the *M-MEM* scores of prompts generalize to some extent across splits—for a given model, the best prompts for a given set of *PER* are likely to work well for a different set.

Table 5 illustrates this by comparing each model's best and worst prompts, showing that the (model-dependent) best and worst prompts for the dev set are still among the best and worst for the test set. These results show that while prompt quality depends on the model, it still generalizes for different names.

**Table 5. Generalization of prompt ranks.** The *PER* entity *M-MEM* scores and corresponding ranks of the best (Rank=1) and worst (Rank=-1) prompts from the dev set on the test set.

| Model | Prompts | dev | | test | |
|---|---|---|---|---|---|
| | | Rank | *M-MEM* | Rank | *M-MEM* |
| ALBERT-B | Bravo, MASK, what an impressive performance! | 1 | 74.84 | 2 | 75.10 |
| | Are you going to MASK's art gallery opening tonight? | -1 | 58.92 | -1 | 58.74 |
| ALBERT-L | Oh, MASK, you're a true gem in our team. | 1 | 73.12 | 3 | 71.38 |
| | MASK, practice forgiveness towards yourself and others. | -1 | 64.60 | -3 | 64.70 |
| BERT-B | Are you going to MASK's art gallery opening tonight? | 1 | 73.96 | 1 | 71.84 |
| | Did MASK give you any advice on starting something new? | -1 | 65.11 | -2 | 63.30 |
| BERT-L | What project is MASK working on? | 1 | 71.56 | 2 | 69.30 |
| | I had a chance to meet MASK's family. | -1 | 67.78 | -1 | 64.42 |
| RoBERTa-B | MASK, can you recommend a good restaurant in town? | 1 | 70.76 | 2 | 70.78 |
| | I had a great conversation with MASK at the party. | -1 | 57.00 | -2 | 59.63 |
| RoBERTa-L | MASK, invest in meaningful relationships. | 1 | 75.27 | 1 | 72.79 |
| | MASK, practice playing the guitar. | -1 | 70.61 | -96 | 70.05 |

## 5.3 Can we further increase the prompt performance gap?

**The prompt set outperforms baselines:** In Table 6, all the *M-MEM* scores achieved by best prompts (B-PT) of the prompt set are higher than the three baselines in the dev set (only a few exceptions in the test set), showing the necessity to robustly detect the model memorization with diverse prompts.

**Prompt engineering outperforms the original prompt set:** We observe that the best/worst-modified prompts (BM-PT and WM-PT) using prompt engineering increase the score gap between the B-PT and the W-PT. For example, the BM-PT of RoBERTa-B improves by approximately 2 percentage points up on the B-PT on the dev set, and the WM-PT decreases the *M-MEM* score of BERT-L by 19 percentage points for the *PER* entity.

**Ensembling techniques do not bring consistent improvement:** The ensembling results of *PER* entity in Table 7 demonstrate that the ensemble techniques (EPT) do not improve/decrease the *M-MEM* scores from the B-PT/W-PT scores. However, we observed a few exceptions for the results of *LOC* and *ORG* entities.

## 5.4 Sentence-level analysis of prompt properties

This section investigates different prompt properties that may influence the *M-MEM* scores for *PER* entity. Analysis of other entity types is shown in S1 Appendix. **Prompt type:** Fig 3 presents prompts' *M-MEM* scores grouped by prompt type. RoBERTa-B has very distinct M-MEM scores for different types: imperative prompts generally yield higher scores than declarative prompts. Conversely, BERT-L has similar *M-MEM* score profiles for different prompt types. Overall, we observe that some models show similar *M-MEM* scores across sentence types, others are more sensitive to particular types.

**Table 6. Results of using different prompt strategies.** One-PT and Mix-PT are baselines adapted from [13]. Bold/italic highlights the highest/lowest values. Δ measures the gap between the highest and lowest values.

| | Prompt Strategy | | ALBERT-B dev | test | ALBERT-L dev | test | BERT-B dev | test | BERT-L dev | test | RoBERTa-B dev | test | RoBERTa-L dev | test |
|---|---|---|---|---|---|---|---|---|---|---|---|---|---|---|
| **PER** | BSL | ∅-PT | 71.11 | 72.44 | 70.81 | 68.91 | 66.78 | 66.10 | 70.16 | 68.11 | 66.39 | 65.46 | 74.40 | 70.29 |
| | | One-PT | 70.42 | 71.18 | 69.02 | 67.57 | 72.15 | 70.31 | 69.91 | 66.49 | 68.57 | 70.72 | 72.68 | 70.22 |
| | | Mix-PT | 68.63 | 70.02 | 67.58 | 66.98 | 70.59 | 69.27 | 69.29 | 66.56 | 66.47 | 67.29 | 72.80 | 70.42 |
| | OPT | B-PT | 74.84 | **75.10** | 73.12 | 71.38 | 73.96 | 71.84 | 71.56 | 69.30 | 70.76 | 70.78 | 75.27 | 72.79 |
| | | W-PT | 58.92 | 58.74 | 64.60 | 64.70 | 65.11 | 63.30 | 67.78 | 64.42 | 57.00 | 59.63 | 70.61 | 70.05 |
| | PTE | BM-PT | 75.14 | 74.96 | 73.14 | 71.44 | 75.14 | 73.31 | 73.03 | 72.18 | 72.68 | 71.59 | 76.28 | 73.33 |
| | | WM-PT | 41.92 | 44.53 | 63.76 | 63.52 | 59.15 | 62.04 | 48.68 | 49.98 | 55.29 | 58.21 | 69.15 | 68.52 |
| | **Δ** | | 33.22 | 30.57 | 9.38 | 7.92 | 15.99 | 11.27 | 24.35 | 22.20 | 17.39 | 13.38 | 7.13 | 4.81 |
| **LOC** | BSL | ∅-PT | 70.06 | 69.15 | 72.32 | 72.83 | 65.70 | 66.62 | 63.31 | 64.31 | 70.34 | 68.53 | 73.18 | 70.69 |
| | | One-PT | 63.35 | 61.35 | 79.02 | 79.70 | 67.55 | 68.06 | 68.76 | 68.67 | 71.55 | 73.62 | 74.18 | 74.81 |
| | | Mix-PT | 69.78 | 68.81 | 76.45 | 76.39 | 68.99 | 68.89 | 71.86 | 70.51 | 73.18 | 75.22 | 73.16 | 72.52 |
| | OPT | B-PT | 81.19 | 80.24 | 82.38 | **81.60** | 73.68 | **72.70** | 78.39 | 77.60 | 78.24 | 80.37 | 79.06 | 80.28 |
| | | W-PT | 56.73 | 55.74 | 67.40 | 69.73 | 66.22 | 66.61 | 63.32 | 63.65 | 65.58 | 67.65 | *69.84* | *68.36* |
| | PTE | BM-PT | **81.39** | **81.22** | **82.40** | 81.38 | 73.01 | 72.51 | **78.11** | **78.41** | **78.29** | **80.66** | **80.06** | **81.01** |
| | | WM-PT | *54.96* | *54.86* | *31.10* | *33.67* | *61.30* | *62.28* | *61.66* | *62.97* | *50.96* | *50.99* | 70.38 | 70.19 |
| | **Δ** | | 26.43 | 26.36 | 51.30 | 47.93 | 12.38 | 10.42 | 16.45 | 15.44 | 27.33 | 29.67 | 10.22 | 12.65 |
| **ORG** | BSL | ∅-PT | 74.38 | 75.88 | 74.06 | 77.54 | 75.90 | 78.58 | 77.58 | 80.47 | 74.77 | 75.07 | 73.46 | 76.28 |
| | | One-PT | 72.57 | 74.49 | 72.00 | 76.13 | 69.88 | 73.53 | 72.92 | 76.00 | 74.19 | 73.99 | 66.72 | 69.09 |
| | | Mix-PT | 68.15 | 68.91 | 69.33 | 72.04 | 69.76 | 72.44 | 69.49 | 71.83 | 68.94 | 68.03 | 66.38 | 68.27 |
| | OPT | B-PT | 76.22 | 76.14 | 75.41 | 77.00 | 78.46 | 79.63 | 79.10 | 79.94 | 77.08 | 76.51 | 74.38 | 75.52 |
| | | W-PT | *56.39* | *58.66* | *56.75* | *60.97* | 63.97 | 67.34 | 65.90 | 68.14 | *56.82* | *57.59* | *58.47* | *60.47* |
| | PTE | BM-PT | **77.21** | **76.87** | **77.31** | **78.19** | **78.92** | **80.05** | **79.55** | **81.63** | **78.17** | **77.46** | **75.50** | **76.49** |
| | | WM-PT | 57.59 | 60.01 | 58.25 | 62.33 | *63.86* | *66.70* | *65.63* | *67.77* | 56.85 | 57.73 | 58.82 | 60.95 |
| | **Δ** | | 20.82 | 18.21 | 20.56 | 17.22 | 15.06 | 13.35 | 13.92 | 13.86 | 21.35 | 19.87 | 17.03 | 16.02 |

**Table 7. Results of using different prompt ensemble strategies.** Bold/italic highlights the scores that are higher/lower than the prompt set (B-PT to W-PT) respectively.

| | Prompt Strategy | ALBERT-B | | ALBERT-L | | BERT-B | | BERT-L | | RoBERTa-B | | RoBERTa-L | |
|---|---|---|---|---|---|---|---|---|---|---|---|---|---|
| | | dev | test | dev | test | dev | test | dev | test | dev | test | dev | test |
| *PER* | MV | 73.24 | 74.30 | 70.96 | 69.71 | 70.75 | 69.34 | 69.91 | 67.98 | 64.87 | 66.30 | 73.73 | 71.05 |
| | AVG-C | 72.81 | 73.22 | 68.20 | 67.43 | 69.66 | 67.95 | 69.91 | 67.95 | 67.76 | 67.67 | 73.97 | 71.16 |
| | WED-C | 72.82 | 73.22 | 68.24 | 67.45 | 69.69 | 68.01 | 69.92 | 67.95 | 67.76 | 67.66 | 73.97 | 71.16 |
| | MAX-C | 71.53 | 73.71 | 70.65 | 68.99 | 73.52 | 71.40 | 69.97 | 67.31 | 61.47 | 63.22 | 73.69 | 71.25 |
| | MIN-C | 60.12 | 59.77 | 64.74 | 65.39 | 68.79 | 66.56 | 68.62 | 67.94 | 68.35 | 68.08 | 73.08 | 71.58 |
| *LOC* | MV | 80.55 | 80.40 | 81.44 | 81.46 | 70.90 | 70.29 | 73.51 | 72.53 | 74.57 | 77.35 | 77.18 | 77.97 |
| | AVG-C | 78.70 | 77.84 | 81.01 | 81.08 | 69.62 | 69.61 | 70.42 | 69.69 | 75.06 | 75.33 | 76.71 | 75.02 |
| | WED-C | 78.90 | 78.12 | 81.03 | 81.09 | 69.68 | 69.65 | 70.60 | 69.85 | 75.12 | 75.38 | 76.75 | 75.06 |
| | MAX-C | 79.21 | **81.02** | 81.87 | 81.51 | **73.98** | **73.80** | 76.61 | 76.24 | 76.92 | 79.11 | 78.96 | 79.81 |
| | MIN-C | 64.53 | 63.08 | 71.57 | 70.03 | *65.16* | *64.52* | *60.70* | *60.73* | 71.06 | 69.08 | *69.44* | *68.33* |
| *ORG* | MV | 73.58 | 75.65 | 70.58 | 74.89 | 72.12 | 75.20 | 74.65 | 77.63 | 71.20 | 72.37 | 66.17 | 68.94 |
| | AVG-C | 72.95 | 75.75 | 71.39 | 75.63 | 72.56 | 75.45 | 74.29 | 76.71 | 69.15 | 70.94 | 70.25 | 72.70 |
| | WED-C | 73.02 | 75.83 | 71.53 | 75.74 | 72.62 | 75.49 | 74.34 | 76.76 | 69.30 | 71.08 | 70.33 | 72.78 |
| | MAX-C | **76.44** | 75.91 | 65.23 | 67.31 | 71.80 | 74.07 | 71.51 | 74.11 | 74.73 | 74.83 | *57.69* | 58.60 |
| | MIN-C | 64.17 | 67.60 | 67.97 | 70.77 | 68.37 | 71.61 | 66.89 | 70.05 | 60.22 | 60.91 | 65.22 | 67.49 |

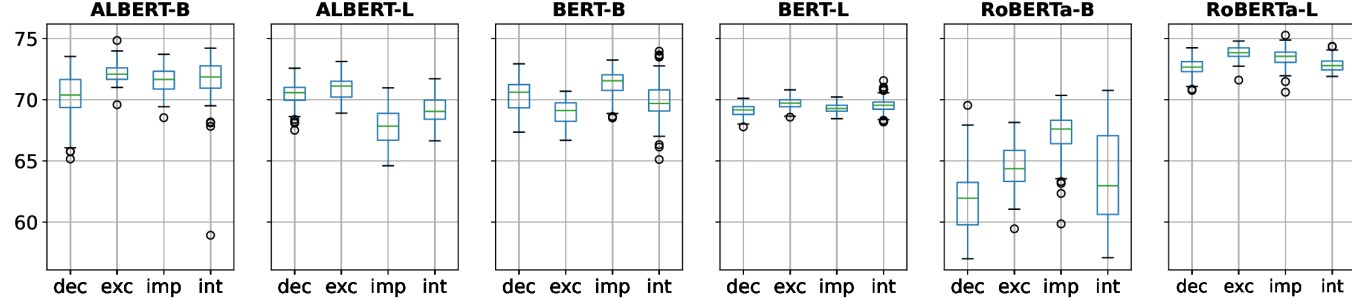

**Fig 3. *M-MEM* scores grouped by prompt type.** The four prompt types are **dec**larative, **exc**lamatory, **imp**erative, and **int**errogative.

**Person's name token position:** Fig 4 shows prompts' *M-MEM* scores grouped by the token position of the name. The closer the name is to the beginning of the prompt, the higher the M-MEM score tends to be for RoBERTa-B and BERT-B.

**Prompt token length:** Fig 5 contrasts prompts' *M-MEM* scores and length (in number of tokens). We find that BERT-B stands out as an outlier, showing that a moderate negative correlation existed between *M-MEM* scores and prompt token lengths. While a weak correlation

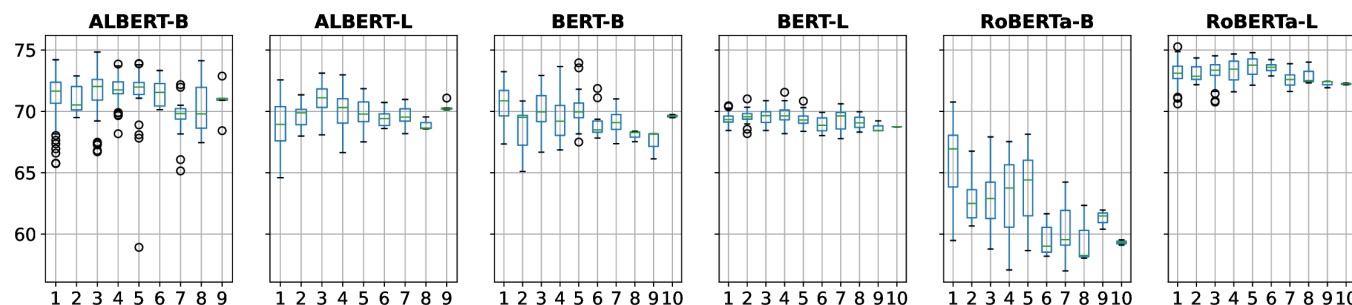

**Fig 4. *M-MEM* scores grouped by person's name token position.**

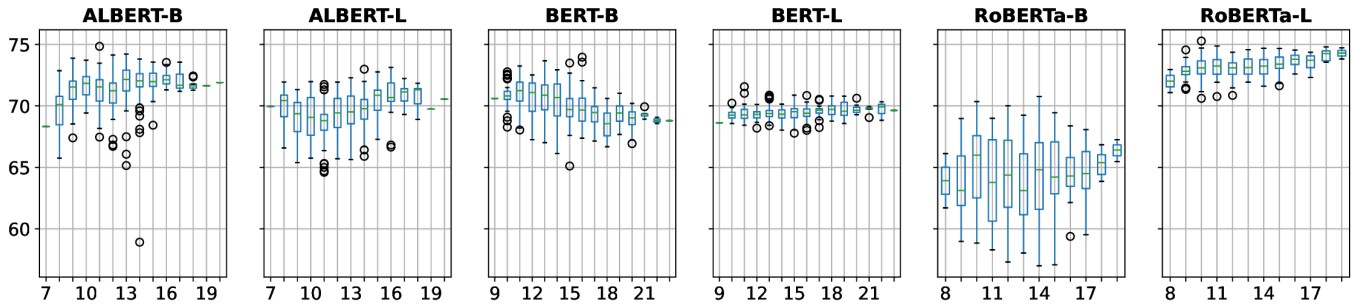

**Fig 5. *M-MEM* scores grouped by prompt token length.**

between *M-MEM* scores and prompt length is observed for most models, meaning that longer prompts are generally more effective at identifying memorization.

## 5.5 Token-level analysis of token importance

Fig 6 shows a token-level analysis of how tokens within a prompt impact the *M-MEM* scores on the example of the BERT-B model for *PER* entity names. Please refer to S1 Appendix for the analysis of other models considered in our experiments.

Individual token importance for a prompt is calculated as the difference between the *M-MEM* scores of the prompt with and without this token on the dev set. These scores are then normalized using a softmax function.

Our results demonstrate that **removing the least important tokens from the prompt increases the *M-MEM* score, while removing the most important tokens has the opposite**

| Prompt | M-MEM | Prompt | M-MEM |
|---|---|---|---|
| Are you going to MASK 's art gallery opening tonight ? | 73.96 | Did MASK give you any advice on starting something new ? | 65.11 |
| Are you going MASK 's art gallery opening tonight ? | 74.65 | Did MASK you any advice on starting something new ? | 61.89 |
| Are you going MASK 's art gallery tonight ? | 74.75 | Did MASK you any on starting something new ? | 60.76 |
| Are you going MASK 's art gallery ? | 74.94 | Did MASK you any on something new ? | 60.34 |
| Are you going MASK 's art gallery | 75.03 | Did MASK you any on something ? | 60.29 |
| you going MASK 's art gallery | 75.14 | Did MASK you on something ? | 61.44 |
| you going MASK 's gallery | 74.70 | Did MASK you something ? | 60.71 |
| going MASK 's gallery | 74.33 | MASK you something ? | 59.15 |
| MASK 's gallery | 73.72 | MASK you something | 61.70 |
| MASK 's | 72.49 | MASK something | 67.69 |
| MASK | 66.78 | MASK | 66.78 |

**Fig 6. Analysis of token importance in prompts.** The figure shows the prompt analysis for the *PER* entity with the *M-MEM* scores on the BERT-B model. The best-performing (left) and the worst-performing (right) prompts were selected on the dev set. The heatmaps are generated with leave-one-token-out: at each step, the least important token is removed from the best-performing prompt, and the most important token is removed from the worst-performing prompt. The normalized token importance scores are under the corresponding tokens. The removed tokens are underlined.

**effect.** For example, iteratively removing the least important tokens from the best-performing prompt for *PER* names (*"Are you going to MASK's art gallery opening tonight?"*) enables an increase in the *M-MEM* score from 73.96 to 75.14 for the prompt *"you going MASK's art gallery"*. Continuing to remove tokens results in a slight decrease in prompt performance, which leads to noun-phrase prompts performing worse than the initial prompt. In the case of the worst-performing prompt, removing the most important token *give* from the initial prompt results in a major decrease in the *M-MEM* score by approximately 3.5 percentage points. Further removal of important tokens slightly reduces the score to a minimum of 59.15 for the prompt *"MASK you something?"*.

**The tokens that function as verbs and the tokens that are located near the *PER* name tokens tend to have a higher importance score.** Tokens like *"going", "give", "practice", "working", "recommend"*, etc. (see Fig 6 and figures in S1 Appendix) have a higher influence on the *M-MEM* score in each considered prompt. A similar trend can be seen for the tokens *"'s", "any", "Oh", "Bravo", "What"*, and *","* located nearer to the *MASK* token (substituted by the *PER* names in the experiments) than tokens located further away.

## 5.6 Self-attention interpretation analysis

Fig 7 shows the attention heatmaps of BERT-B for the best and the worst prompts with each prompt completed with in-sample and out-of-sample *PER* names separately.

Generally, we notice that attention tends to be distributed similarly given a prompt, focusing on the first and the entity name tokens. This observation applies to all analyzed models (heatmaps for the other five models are shown in S1 Appendix).

**Name level analysis:** We observe a slightly higher focus on in-sample entity names than out-of-sample entity names when comparing the averaged attention weights on both the best and worst prompts.

**Prompt level analysis:** When comparing the heatmaps in the prompt level, we notice that models tend to focus more (or equally for BERT-B shown in Fig 7) on the entity name tokens in the best prompt than the worst, except for when the entity name tokens in the worst prompt are positioned in the beginning the prompt.

## 6 Discussion

Through our comprehensive analysis, we acknowledge the following questions for discussion.

**How much memorization poses an unacceptable risk?** While our study quantifies memorization based on a model's differential confidence between in-sample and out-of-sample

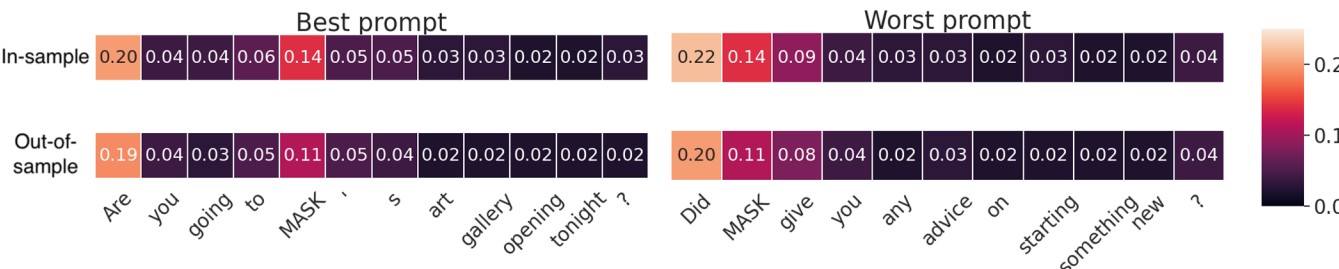

**Fig 7. Attention heatmaps of the best and the worst prompts.** The figure shows the result of BERT-B for *PER* entity names on the dev set averaged over all attention heads and layers. The attention weights corresponding to each prompt's "MASK" token are used and averaged over the in-sample and out-of-sample *PER* entity names separately.

entity names, we do not define a threshold for what constitutes an "unacceptable" level of memorization. This is because the determination of the level is highly application-dependent. For example, even minimal memorization may be intolerable in domains involving sensitive personal information, such as healthcare or finance (e.g., HIPAA compliance or banking privacy) [20,32], or in contexts involving intellectual property and copyrighted material [33,34]. However, more leniency may be acceptable in tasks involving public or non-sensitive data [35]. Defining acceptable risk levels in practice typically requires considering domain-specific factors, regulatory standards, and ethical judgments. We acknowledge this as an important area for future interdisciplinary research.

**How do we weigh risk versus performance in determining acceptable risk in various contexts?** In this work, we propose a method to measure memorization in NER models by comparing prediction confidences for in-sample and out-of-sample entity names. While this quantification is a necessary step toward understanding privacy risks, determining what constitutes an "acceptable" level of memorization remains an open and context-dependent question. The trade-off between performance and privacy varies across applications and domains. Thus, we do not attempt to define this threshold here; instead, we highlight that our measurement framework provides the basis for such judgments. A crucial next step is to develop and evaluate mitigation strategies, such as regularization techniques, differential privacy, or selective data obfuscation, that can reduce memorization without substantially degrading model performance [36].

**How to interpret M-MEM scores in context?** The M-MEM score offers a way to quantify and compare memorization-related risk in NER models by measuring how consistently a model favors in-sample entity names over out-of-sample alternatives. While this metric provides valuable insights into model behavior, interpreting it as a measure of privacy risk depends on the application context. For example:

- High M-MEM + sensitive data (e.g., patient names): likely an unacceptable privacy risk.
- High M-MEM + public data (e.g., Wikipedia): possibly tolerable depending on usage.
- Low M-MEM: indicates weaker memorization, generally favorable from a privacy standpoint.

M-MEM does not directly estimate the probability of a successful attack or legal threshold violations; rather, it serves as a comparative indicator of how memorized a model is with respect to its training entities. Future work can refine this score into a more interpretable privacy metric or combine it with attack success rates to estimate risk more holistically [37].

## 7 Limitations

We address the limitations of our work as follows:

- This research focused on MLM-based models. Therefore, the analysis and findings in this paper may not be generalizable to other types of models, such as auto-regressive models, in the same way as previous studies of memorization in auto-regressive models are not generalizable to MLMs. Our work fills a research gap of examining memorization in fine-tuned MLMs.
- The models that we investigated had been trained by other researchers and the snapshots at different training stages (as well as the complete training settings) are unavailable to us. Thus, we can not provide dynamic memorization analysis across training procedures.
- We opted to study widely-used state-of-the-art models, in a setting where all components (base models, training data, etc) are widely available rather than non-SOTA models, or

models trained for specific domains. Unfortunately, those models are overwhelmingly fine-tuned on the CoNLL-2003 dataset, the predominant benchmark for NER. However, the CoNLL-2003 dataset is sampled from the Reuters Corpus (a corpus consisting of Reuters news stories), which contains rich and diverse topics such as sports, business, and politics. Nevertheless, our experiments across three distinct entity types enhance the generalizability of our findings.

## 8 Conclusion

We studied memorization in fine-tuned MLM-based NER models. Unlike auto-regressive language models, where memorization can be assessed by prompting the model to generate verbatim segments of its training data [3,20], the output of NER models consists of structured label predictions (e.g., B-PER, I-ORG), and thus does not support such evaluation. Our study quantifies memorization of NER models without relying on generative capacity. In this setting, we compared a fixed set of five hand-written prompts to 400 automatically generated prompts for each entity type, and we measured how well those prompts could distinguish entity names that were presented in the fine-tuning data from those that were not, using a large set of candidate entity names.

A detailed analysis revealed that a prompt's ability to detect memorization varies between models but remains stable across different entity sets. We find a large variability in the performance of generated prompts, and that many generated prompts significantly outperform hand-written ones. The effectiveness of prompts can be increased by removing the least important tokens even if the prompts become ungrammatical after the removal. Overall, we demonstrate the importance of using diverse prompts through comprehensive experiments and analysis. We provide all the prompts in supplemental material to detect memorization of NERs for future privacy studies.

## Supporting information

**S1 Appendix.**
(PDF)

**S2 Dataset.**
(TXT)

**S3 Code.** (**GitHub repository:** https://github.com/Yuuxii/NER-memorization-detection/releases/tag/v1.0).

## Acknowledgments

We thank Vasisht Duddu and N. Asokan who dedicated time and expertise to provide valuable feedback on this paper.

## Author contributions

**Conceptualization:** Yuxi Xia, Lisa Nußbaumer, Benjamin Roth.

**Data curation:** Yuxi Xia, Lisa Nußbaumer.

**Formal analysis:** Yuxi Xia, Anastasiia Sedova, Pedro Henrique Luz de Araujo, Vasiliki Kougia, Lisa Nußbaumer.

**Funding acquisition:** Benjamin Roth.

**Investigation:** Yuxi Xia, Anastasiia Sedova, Pedro Henrique Luz de Araujo, Vasiliki Kougia, Lisa Nußbaumer.

**Methodology:** Yuxi Xia, Benjamin Roth.

**Project administration:** Yuxi Xia.

**Resources:** Benjamin Roth.

**Software:** Yuxi Xia.

**Supervision:** Benjamin Roth.

**Validation:** Yuxi Xia.

**Visualization:** Yuxi Xia, Anastasiia Sedova, Pedro Henrique Luz de Araujo, Vasiliki Kougia.

**Writing – original draft:** Yuxi Xia, Anastasiia Sedova, Pedro Henrique Luz de Araujo, Vasiliki Kougia.

**Writing – review & editing:** Yuxi Xia, Anastasiia Sedova, Pedro Henrique Luz de Araujo, Vasiliki Kougia, Benjamin Roth.

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
