## [Decision Letter · Decision Letter 0]

13 May 2025

PONE-D-25-05072Exploring prompts to elicit memorization in masked language model-based named entity recognitionPLOS ONE

Dear Dr. Xia,

Thank you for submitting your manuscript to PLOS ONE. After careful consideration, we feel that it has merit but does not fully meet PLOS ONE’s publication criteria as it currently stands. Therefore, we invite you to submit a revised version of the manuscript that addresses the points raised during the review process.

We look forward to receiving your revised manuscript.

Kind regards,

Thiago P. Fernandes, PhD

Academic Editor

PLOS ONE

Journal Requirements:

“This research has been funded by the Vienna Science and Technology Fund (WWTF) [10.47379/VRG19008] “Knowledge-infused Deep Learning for Natural Language Processing”.”

3. Please note that your Data Availability Statement is currently missing the repository name. If your manuscript is accepted for publication, you will be asked to provide these details on a very short timeline. We therefore suggest that you provide this information now, though we will not hold up the peer review process if you are unable.

4. We are unable to open your Supporting Information file [supporting.zip]. Please kindly revise as necessary and re-upload.

**Additional Editor Comments:**

Please respond all comments and highlight them in the version of the ms.

Reviewers' comments:

Reviewer's Responses to Questions

**Comments to the Author**

1. Is the manuscript technically sound, and do the data support the conclusions?

Reviewer #1: Yes

Reviewer #2: Partly

2. Has the statistical analysis been performed appropriately and rigorously? 

Reviewer #1: Yes

Reviewer #2: No

3. Have the authors made all data underlying the findings in their manuscript fully available?

Reviewer #1: Yes

Reviewer #2: Yes

4. Is the manuscript presented in an intelligible fashion and written in standard English?

Reviewer #1: Yes

Reviewer #2: Yes

5. Review Comments to the Author

Reviewer #1: I think the content of the manuscript is novel and appropriate, but the structure could be improved. Specifically,

1. Throughout the manuscript, both generalized and model-specific findings are reported. I believe the findings would be easier to follow and refer back to (and possibly be more impactful) if either all the model-specific findings were reported first, then all the generalized findings (or if each sub-section reported all the model-specific findings first followed by all the generalized findings for that sub-section).

2. More precise definition of “memorization”. For example, reference 3 in this version of the manuscript includes

> Definition 3.1. A string s is extractable with k tokens of context from a model f if there exists a (length-k) string p, such that the concatenation [p || s] is contained in the training data for f, and f produces s when prompted with p using greedy decoding.

followed by two paragraphs contrasting this definition with other definitions of the term. If the definition used here is the same as one used in one of the references, a simple “as defined in [reference]” would be perfectly clarifying.

3. I think answering these questions is beyond the scope of this manuscript, but the questions themselves should be acknowledged:

a. How much memorization poses an unacceptable risk?

b. How do we weigh risk versus performance in determining acceptable risk in various contexts?

This work can be an answer to “how do we compare the memorization-related risk of a given NER model?” but why we’re measuring memorization and how to interpret the M-MEM scores relative to the risk could be clearer.

4. The very end of the conclusion (lines 352‒354) states, “Overall, we demonstrate the importance of using diverse prompts (or generating target prompts by following guidance in Sections 5.4 and 5.5) to detect memorization of NERs for future privacy studies with a comprehensive analysis.” Rather than recommending readers refer to the middle of the Results and Analysis section of this paper, the guidance could be presented as a step-by-step guide (either as its own (sub-section) in the manuscript or in an online resource or a supplemental document). Having the information in a paper like this is useful, but if we want people who read the paper to follow the guidance, we should reduce the friction to doing so (by making it easy to refer to the guidance in an isolated way).

5. Some editorial pedantry:

a. Lines 6‒8: I think you mean, “While training data can be extracted through direct prompt querying and text generation in GPT models, that method is not applicable to Masked Language Models (MLMs).” (“that method” instead of “a method that”)

b. Line 48: I don’t think we want the word “only” here. This work shows me that prompt engineering can increase the performance gap, but not that nothing else can.

c. Line 323: I think “Our work fills a research gap” rather than “Our work fills the research gap” is more appropriate; the work does fill a gap, but “fills the […] gap” implies there’s no need for further research here.

d. Lines 343‒344: I think you mean “400 automatically generated prompts” instead of “automatically generated 400 prompts.”

Reviewer #2: Summary:

This paper studies named-entity recognition (NER) in masked language models (MLM) and assesses whether models finetuned on NER datasets will be more biased toward named-entities seen during finetuning (referred to as memorization in the paper). The paper focuses on how different types of linguistic contexts (referred to as prompts in the paper) affect the surprisal scores models assign to entity tokens. Specifically, the authors collect a large sample of linguistic contexts using an LLM and construct NER examples by inserting a controlled set of named-entities (ones seen vs. unseen during finetuning) into the context sentences. The results show that model surprisal scores differ substantially with respect to the context sentence used, supporting an argument that a diverse set of contexts should be used when evaluating models’ abilities to memorize text.

Strengths:

- Results show that different contexts used to prompt the model can substantially affect the amount of memorization reflected in model outputs, raising the importance of trying different contexts when assessing a model’s ability to memorize text.

- A diverse set of experiments have been performed to assess the effect of linguistic context on memorization of named-entities.

Weaknesses:

- The paper presents its novelty by claiming that masked language models cannot perform the same type of evaluations as autoregressive language models (LMs) because MLM’s “inability to generate text”. However, it is possible to generate text using MLMs and refutes the authors' claims in several places in the paper (line 12 and 341).

- The choice of the evaluation metric in Equation 1 needs to be justified (see questions for more details). Under the current formulation, the metric is inherently biased toward seen entities. The validity of the experimental results is in question because this metric is relevant in all subsequent experiments.

- The authors chose to use non-standard machine learning terminologies in the paper that makes it unnecessarily confusing to read and can potentially mislead the reader on what the contributions are. Specifically, the following terminologies seem to deviate from their standard use from ML/NLP literature:

1) ‘in-train’ and ‘out-of-train’: It is more standard to refer to such examples as ‘in-sample’ and ‘out-of-sample’. Alternatively, they can be referred to as ‘seen’ and ‘unseen’ examples.

2) What the authors have been referring to as ‘prompts’ in most of the paper are really ‘linguistic contexts’ or ‘context sentences’. Prompt usually refers to the language used to describe a specific task, but in this paper, the models are evaluated against different linguistic contexts that contain certain types of named entities. In other words, it is the contexts that are different when two different ‘prompts’ are compared, not the task description. Furthermore, the authors referred to one of their experimental conditions as ‘prompt engineering’ (line 180) which further exacerbates the abuse of terminology.

3) The term ‘memorization’ in the context of language modeling typically refers to the case where an LM outputs parts of its training text verbatim during inference time. In the case of NER specifically, memorization would refer to the case where the model reproduces, or assigns high probability scores for, the exact pair of linguistic context and named-entity. Under the paper’s setup, however, the context is not necessarily paired with the inserted named-entity during training/finetuning, so I would refer to what the authors are measuring as “model bias towards seen named-entities” rather than memorization.

Questions/comments:

- Line 12 and 341. Why are MLM-based models unable to generate text? MLMs work differently from autoregressive LMs but can still be used to perform text generation. These statements should be clarified.

- What’s so special about MLMs that sets them apart from autoregressive LMs when it comes to memorization? Is there something special about your experimental approach that differentiates it from previous approaches applied to autoregressive LMs?

- In equation 1, what’s the rationale behind taking the maximum probability score of an entity’s beginning (B-e) and inside (I-e) tokens instead of taking the average? Under the current formulation, as long as one of the inside tokens attains a high probability score, that specific entity would receive a high score. This formulation would cause a bias towards seen (i.e., in-train) entities. For example, a model may assign equally low probabilities to the first token of a seen entity and an unseen entity (which suggests equal surprisal), but as soon as you condition on the first token, the model would assign much higher likelihood scores to the inside tokens of seen entities because they’ve been seen during finetuning. Under the current formulation, therefore, the seen entity would receive much higher scores despite the model being equally surprised when the first token of the entity is evoked.

Typos & Suggestions:

- Line 7. Broken sentence.

- Line 76 and 342. ‘previous works’ -> ‘previous work’

- Line 114. Extra number?

- Line 123. “In-train names are randomly sampled from Dout for each entity type”. Do you mean out-train here? Otherwise, you are sampling training examples from the unseen name set?

- Line 339. ‘pre-training or’ -> ‘pre-training of’

- Line 343. ‘generated’ -> ‘generate’

6. PLOS authors have the option to publish the peer review history of their article (what does this mean?). If published, this will include your full peer review and any attached files.

Reviewer #1: **Yes: **Jon Cluce

Reviewer #2: No

---

## [Author Response · Author response to Decision Letter 1]

23 Jun 2025

Dear Editors and Reviewers,

We thank the reviewers for their thoughtful and constructive feedback on our manuscript, titled "Exploring Prompts to Elicit Memorization in Masked Language Model-based Named Entity Recognition" (Manuscript ID: [PONE-D-25-05072]). We greatly appreciate the time and effort dedicated to evaluating our work, and we have carefully considered each comment in preparing our revised manuscript.

Below, we provide a point-by-point response to each reviewer’s comment. We believe that the revisions have significantly improved the quality, clarity, and impact of the paper, and we hope the updated version addresses the reviewers’ concerns satisfactorily

Reviewer #1: I think the content of the manuscript is novel and appropriate, but the structure could be improved. Specifically,

1. Throughout the manuscript, both generalized and model-specific findings are reported. I believe the findings would be easier to follow and refer back to (and possibly be more impactful) if either all the model-specific findings were reported first, then all the generalized findings (or if each sub-section reported all the model-specific findings first followed by all the generalized findings for that sub-section).

Response: We address the comment by reporting the model-specific findings first and then reporting the generalized findings in Sections 5.1, 5.2, and 5.4.

2. More precise definition of “memorization”. For example, reference 3 in this version of the manuscript includes

> Definition 3.1. A string s is extractable with k tokens of context from a model f if there exists a (length-k) string p, such that the concatenation [p || s] is contained in the training data for f, and f produces s when prompted with p using greedy decoding.

followed by two paragraphs contrasting this definition with other definitions of the term. If the definition used here is the same as one used in one of the references, a simple “as defined in [reference]” would be perfectly clarifying.

Response: We clarified our definition of memorization in Section 3.3 by adding Definition 3.1.

3. I think answering these questions is beyond the scope of this manuscript, but the questions themselves should be acknowledged:

a. How much memorization poses an unacceptable risk?

b. How do we weigh risk versus performance in determining acceptable risk in various contexts?

This work can be an answer to “how do we compare the memorization-related risk of a given NER model?” but why we’re measuring memorization and how to interpret the M-MEM scores relative to the risk could be clearer.

Response: We addressed the two questions mentioned above and the interpretation of the M-MEM scores in Section 6, Discussion.

4. The very end of the conclusion (lines 352‒354) states, “Overall, we demonstrate the importance of using diverse prompts (or generating target prompts by following guidance in Sections 5.4 and 5.5) to detect memorization of NERs for future privacy studies with a comprehensive analysis.” Rather than recommending readers refer to the middle of the Results and Analysis section of this paper, the guidance could be presented as a step-by-step guide (either as its own (sub-section) in the manuscript or in an online resource or a supplemental document). Having the information in a paper like this is useful, but if we want people who read the paper to follow the guidance, we should reduce the friction to doing so (by making it easy to refer to the guidance in an isolated way).

Response: We changed the corresponding sentence in the Conclusion section, as we provide all the prompts and implementation code as a supplemental document.

5. Some editorial pedantry:

a. Lines 6‒8: I think you mean, “While training data can be extracted through direct prompt querying and text generation in GPT models, that method is not applicable to Masked Language Models (MLMs).” (“that method” instead of “a method that”)

b. Line 48: I don’t think we want the word “only” here. This work shows me that prompt engineering can increase the performance gap, but not that nothing else can.

c. Line 323: I think “Our work fills a research gap” rather than “Our work fills the research gap” is more appropriate; the work does fill a gap, but “fills the […] gap” implies there’s no need for further research here.

d. Lines 343‒344: I think you mean “400 automatically generated prompts” instead of “automatically generated 400 prompts.”

Response: We changed the manuscript according to the above points.

Reviewer #2::

- The paper presents its novelty by claiming that masked language models cannot perform the same type of evaluations as autoregressive language models (LMs) because MLM’s “inability to generate text”. However, it is possible to generate text using MLMs and refutes the authors' claims in several places in the paper (line 12 and 341).

Response: We acknowledge that MLMs can generate text, but less naturally or sequentially than autoregressive LMs (e.g., GPT). Techniques such as Gibbs sampling have been used to generate text with pre-trained MLMs [1]. However, this can not be applied to fine-tuned MLMs for NER, because the token reconstruction head used during pre-training of MLM is replaced with a task-specific classification head which is restricted to the target space, e.g., named entity labels [2, 3]. This makes standard generation-based memorization probing inapplicable to NER models. We address this point by revising the sentence on Line 12 “inability to generate text” to “During fine-tuning for NER, the model's objective shifts entirely to sequence labeling, and the token reconstruction head used during pre-training is replaced with a task-specific classification head. This new output layer predicts only NER tags and no longer retains the capability to generate raw text. Hence, NER models cannot be directly assessed for memorization via text generation, such as prompting them to reproduce verbatim in-sample entities given a prefix, a strategy commonly applied to auto-regressive models”. We also extend this in the Related Work and Conclusion (original line 341, but now Line 417) Sections.

[1] Wang A, Cho K. BERT has a Mouth, and It Must Speak: BERT as a Markov Random Field Language Model. In: Bosselut A, Celikyilmaz A, Ghazvininejad M, Iyer S, Khandelwal U, Rashkin H, et al., editors. Proceedings of the Workshop on Methods for Optimizing and Evaluating Neural Language Generation. Minneapolis, Minnesota: Association for Computational Linguistics; 2019.

[2] Yang Z, Ding M, Guo Y, Lv Q, Tang J. Parameter-Efficient Tuning Makes Good Classification Head. In: Goldberg Y, Kozareva Z, Zhang Y, editors. Proceedings of the 2022 Conference on Empirical Methods in Natural Language Processing. Abu Dhabi, United Arab Emirates: Association for Computational Linguistics; 2022. p. 7576–7586

[3] Keraghel I, Morbieu S, Nadif M. Recent Advances in Named Entity Recognition: A Comprehensive Survey and Comparative Study; 2024.

- The choice of the evaluation metric in Equation 1 needs to be justified (see questions for more details). Under the current formulation, the metric is inherently biased toward seen entities. The validity of the experimental results is in question because this metric is relevant in all subsequent experiments.

Response: We explained the reviewer’s confusion about the formula in the question below.

- The authors chose to use non-standard machine learning terminologies in the paper that makes it unnecessarily confusing to read and can potentially mislead the reader on what the contributions are. Specifically, the following terminologies seem to deviate from their standard use from ML/NLP literature:

1) ‘in-train’ and ‘out-of-train’: It is more standard to refer to such examples as ‘in-sample’ and ‘out-of-sample’. Alternatively, they can be referred to as ‘seen’ and ‘unseen’ examples.

Response: We address this comment by replacing in-train and out-of-train with “in-sample” and “out-of-sample”.

2) What the authors have been referring to as ‘prompts’ in most of the paper are really ‘linguistic contexts’ or ‘context sentences’. Prompt usually refers to the language used to describe a specific task, but in this paper, the models are evaluated against different linguistic contexts that contain certain types of named entities. In other words, it is the contexts that are different when two different ‘prompts’ are compared, not the task description. Furthermore, the authors referred to one of their experimental conditions as ‘prompt engineering’ (line 180) which further exacerbates the abuse of terminology.

Response: While we acknowledge that “prompt” can refer to task instructions in autoregressive models, in the context of masked language models, it has also been widely used to describe input templates used for probing or prediction [1, 2, 3]. We follow this precedent in referring to our sentence-level templates as prompts. We added these papers as citations in our paper for clarification (Line 20).

[1] Jiang, Z., Xu, F. F., Araki, J., & Neubig, G. (2020). How can we know what language models know? Zhengbao Jiang, Frank F. Xu, Jun Araki, Graham Neubig; How Can We Know What Language Models Know?. Transactions of the Association for Computational Linguistics 2020; 8 423–438. doi: https://doi.org/10.1162/tacl_a_00324

→ They automatically construct sentence-level prompts to evaluate factual knowledge in masked LMs.

[2] Pengfei Liu, Weizhe Yuan, Jinlan Fu, Zhengbao Jiang, Hiroaki Hayashi, and Graham Neubig. 2023. Pre-train, Prompt, and Predict: A Systematic Survey of Prompting Methods in Natural Language Processing. ACM Comput. Surv. 55, 9, Article 195 (September 2023), 35 pages. https://doi.org/10.1145/3560815

→ This comprehensive survey discusses various prompting techniques across different language models, including the use of prompts as input templates for MLMs to perform specific tasks.

[3] Ziyang Xu, Keqin Peng, Liang Ding, Dacheng Tao, and Xiliang Lu. 2024. Take Care of Your Prompt Bias! Investigating and Mitigating Prompt Bias in Factual Knowledge Extraction. In Proceedings of the 2024 Joint International Conference on Computational Linguistics, Language Resources and Evaluation (LREC-COLING 2024), pages 15552–15565, Torino, Italia. ELRA and ICCL.

→ This paper provides a pertinent example of using the term "prompt" to refer to sentence-level templates in the context of masked language models (MLMs).

3) The term ‘memorization’ in the context of language modeling typically refers to the case where an LM outputs parts of its training text verbatim during inference time. In the case of NER specifically, memorization would refer to the case where the model reproduces, or assigns high probability scores for, the exact pair of linguistic context and named-entity. Under the paper’s setup, however, the context is not necessarily paired with the inserted named-entity during training/finetuning, so I would refer to what the authors are measuring as “model bias towards seen named-entities” rather than memorization.

Response: We acknowledge that in generative language modeling, "memorization" typically refers to a model reproducing verbatim sequences from its training data during free-text generation [1]. In our setup, NER models are not evaluated on reproducing exact entities. Rather, we assess whether the model assigns systematically higher confidence to entities seen during training (in-sample entities) than out-of-sample ones. We use the term memorization in alignment with prior works that extend this notion beyond verbatim reproduction to include biased confidence or likelihood assignment toward seen inputs in structured prediction tasks [2, 3]. In our work, this manifests as a model that assigns higher probabilities to in-sample entities, which we argue constitutes a privacy-relevant form of memorization in NER models.

We have revised the manuscript to explicitly add clarification to avoid confusion with the generative case. (Definition 3.1)

[1] Carlini N, Tramer F, Wallace E, Jagielski M, Herbert-Voss A, Lee K, et al. Extracting training data from large language models. In: 30th USENIX Security Symposium (USENIX Security 21); 2021. p. 2633–2650.

[2] Carlini N, Liu C, ´Ulfar Erlingsson, Kos J, Song D. The Secret Sharer: Evaluating and Testing Unintended Memorization in Neural Networks. USENIX Security Symposium. 2019

[3] Ali RS, Zhao BZH, Asghar HJ, Nguyen T, Wood ID, Kaafar D. Unintended Memorization and Timing Attacks in Named Entity Recognition Models. In: Proceedings on Privacy Enhancing Technologies. PETS 2023; 2023. Available from: http://arxiv.org/abs/2211.02245.

Questions/comments:

- Line 12 and 341. Why are MLM-based models unable to generate text? MLMs work differently from autoregressive LMs but can still be used to perform text generation. These statements should be clarified.

Response: We clarified this comment above.

- What’s so special about MLMs that sets them apart from autoregressive LMs when it comes to memorization? Is there something special about your experimental approach that differentiates it from previous approaches applied to autoregressive LMs?

Response: Auto-regressive language models (e.g., GPT models) can be directly probed for memorization by prompting them to generate text given a prefix. If a model reproduces long sequences seen during training, especially verbatim, this is a strong signal of memorization. Thus, memorization in autoregressive models is typically studied through generation-based evaluations.

However, this can not be applied to fine-tuned MLMs for NER, because the token reconstruction head used during pre-training of MLMs is replaced with a task-specific classification head which is restricted to the target space, e.g., named entity labels. This makes standard generation-based memorization probing inapplicable to NERs.

What’s unique about our experimental approach:

Our approach is tailored to evaluate memorization within the fine-tuned MLM framework, particularly in NER. Instead of prompting the model to generate text, we:

(1) Create diverse prompt templates (contextual sentences) in which candidate named entities (in-sample vs. out-of-sample) are inserted.

(2) Measure prediction confidence scores on the [MASK] token for both in-sample and out-of-sample entities.

(3) Define memorization as the model’s preference for training entities (higher confidence), even when inserted into novel linguistic contexts.

This allows us to quantify memorization behavior in a structured prediction task, without relying on generative capacity. While previous works have examined memorization in autoregressive models largely for text generation tasks, our work is one of the first to provide a systematic and quantifiable approach to studying memorization in fine-tuned MLMs.

We added this part in the Introduction, Related Work and Conclusion Sections to further distinguish our work from methods used for the autoregressive models.

- In equation 1, what’s the rationale behind taking the maximum probability score of an entity’s beginning (B-e) and inside (I-e) tokens instead of taking the average? Under the current formulation, as long as one of the inside tokens attains a high probability score, that specific entity would receive a high score. This formulation would cause a bias towards seen (i.e., in-train) entities. For example, a model may assign equally low probabilities to the first token of a seen entity and an unseen entity (which suggests equal surprisal), but as soon as you condition on the first token, the model would assign much higher likelihood scores to the inside tokens of seen entities because they’ve been seen during finetuning. Under the current formulation, therefore, the seen entity would receive much higher scores despite the model being equally surprised when the first token of the entity is evoked.

Response: We appreciate the reviewer’s careful attention to Equation 1 and the thoughtful critique. However, we believe there may be a misunderstanding regarding how the entity-level confidence score is computed.

To clarify: when evaluating each token in a target entity, the NER model outputs probabilities over all possible labels (e.g., B-PER, I-PER for person entities). Since

---

## [Decision Letter · Decision Letter 1]

23 Jul 2025

PONE-D-25-05072R1Exploring prompts to elicit memorization in masked language model-based named entity recognitionPLOS ONE

Dear Dr. Xia,

Thank you for submitting your manuscript to PLOS ONE. After careful consideration, we feel that it has merit but does not fully meet PLOS ONE’s publication criteria as it currently stands. Therefore, we invite you to submit a revised version of the manuscript that addresses the points raised during the review process.

**Please respond all comments and highlight the changes in the revised ms.**

We look forward to receiving your revised manuscript.

Kind regards,

Thiago P. Fernandes, PhD

Academic Editor

PLOS ONE

**Journal Requirements:**

Reviewers' comments:

Reviewer's Responses to Questions

**Comments to the Author**

1. If the authors have adequately addressed your comments raised in a previous round of review and you feel that this manuscript is now acceptable for publication, you may indicate that here to bypass the “Comments to the Author” section, enter your conflict of interest statement in the “Confidential to Editor” section, and submit your "Accept" recommendation.

Reviewer #1: All comments have been addressed

Reviewer #2: All comments have been addressed

2. Is the manuscript technically sound, and do the data support the conclusions?

Reviewer #1: Yes

Reviewer #2: Yes

3. Has the statistical analysis been performed appropriately and rigorously? 

Reviewer #1: Yes

Reviewer #2: Yes

4. Have the authors made all data underlying the findings in their manuscript fully available?

Reviewer #1: Yes

Reviewer #2: Yes

5. Is the manuscript presented in an intelligible fashion and written in standard English?

Reviewer #1: Yes

Reviewer #2: Yes

6. Review Comments to the Author

**Reviewer #1: **Thank you for addressing the concerns I raised in my review. I'm satisfied with this revision, with a couple tiny revisions:

1. The GitHub link currently points the "main" branch, which is a moving target. Please either create a tag to point to or point to an SHA so the paper will reference a static version of the code.

2a. Reference 12 has subscript "a0" in place of "_a_0" in the DOI.

2b. Some references use DOIs and others use HTTP links where DOIs are available, and some don't have links despite being available online. I suggest using DOIs where available, and HTTP links where DOI links aren't available but the source is available online. This recommendation is very soft; I don't think this should prevent publication.

**Reviewer #2:** (No Response)

7. PLOS authors have the option to publish the peer review history of their article (what does this mean?). If published, this will include your full peer review and any attached files.

Reviewer #1: **Yes: **Jon Cluce

Reviewer #2: No

---

## [Author Response · Author response to Decision Letter 2]

5 Aug 2025

Dear Editors and Reviewers,

We thank the reviewers for their thoughtful and constructive feedback on our manuscript, titled "Exploring Prompts to Elicit Memorization in Masked Language Model-based Named Entity Recognition" (Manuscript ID: [PONE-D-25-05072]). We greatly appreciate the time and effort dedicated to evaluating our work, and we have carefully considered each comment in preparing our revised manuscript.

Below, we provide a point-by-point response to each reviewer’s comment. We believe that the revisions have significantly improved the quality, clarity, and impact of the paper, and we hope the updated version addresses the reviewers’ concerns satisfactorily

1. The GitHub link currently points the "main" branch, which is a moving target. Please either create a tag to point to or point to an SHA so the paper will reference a static version of the code.

Response: We have created a tag named v1.0 and updated the GitHub link in the paper to point to a static version of the code.

2a. Reference 12 has subscript "a0" in place of "_a_0" in the DOI.

Response: We have revised "a0" to "_a_0".

2b. Some references use DOIs and others use HTTP links where DOIs are available, and some don't have links despite being available online. I suggest using DOIs where available, and HTTP links where DOI links aren't available but the source is available online. This recommendation is very soft; I don't think this should prevent publication.

Response: We have revised the references by adding necessary DOIs or HTTP links for each referred paper.

Sincerely,

Yuxi Xia

---

## [Decision Letter · Decision Letter 2]

7 Aug 2025

Exploring prompts to elicit memorization in masked language model-based named entity recognition

PONE-D-25-05072R2

Dear Dr. Xia,

We’re pleased to inform you that your manuscript has been judged scientifically suitable for publication and will be formally accepted for publication once it meets all outstanding technical requirements.

Kind regards,

Thiago P. Fernandes, PhD

Academic Editor

PLOS ONE

Additional Editor Comments (optional):

Reviewers' comments:

Reviewer's Responses to Questions

**Comments to the Author**

1. If the authors have adequately addressed your comments raised in a previous round of review and you feel that this manuscript is now acceptable for publication, you may indicate that here to bypass the “Comments to the Author” section, enter your conflict of interest statement in the “Confidential to Editor” section, and submit your "Accept" recommendation.

Reviewer #1: All comments have been addressed

2. Is the manuscript technically sound, and do the data support the conclusions?

Reviewer #1: Yes

3. Has the statistical analysis been performed appropriately and rigorously? 

Reviewer #1: Yes

4. Have the authors made all data underlying the findings in their manuscript fully available?

Reviewer #1: Yes

5. Is the manuscript presented in an intelligible fashion and written in standard English?

Reviewer #1: Yes

6. Review Comments to the Author

Reviewer #1: Thank you for addressing my final comments. I am satisfied with this version of the manuscript. I think there's still some weirdness in the formatting of some of the references, but editorial should sort that out prior to publication.

7. PLOS authors have the option to publish the peer review history of their article (what does this mean?). If published, this will include your full peer review and any attached files.

Reviewer #1: **Yes: **Jon Cluce

---

## [Editor Report · Acceptance letter]

PONE-D-25-05072R2

PLOS ONE

Dear Dr. Xia,

I'm pleased to inform you that your manuscript has been deemed suitable for publication in PLOS ONE. Congratulations! Your manuscript is now being handed over to our production team.

Kind regards,

on behalf of

Dr. Thiago P. Fernandes

Academic Editor

PLOS ONE